# The Utility of Miniaturized Adsorbers in Exploring the Cellular and Molecular Effects of Blood Purification: A Pilot Study with a Focus on Immunoadsorption in Multiple Sclerosis

**DOI:** 10.3390/ijms25052590

**Published:** 2024-02-23

**Authors:** Andreas Körtge, Anne Breitrück, Sandra Doß, Jacqueline Hofrichter, Sophie-Charlotte Nelz, Horst Krüsemann, Reinhold Wasserkort, Brit Fitzner, Michael Hecker, Steffen Mitzner, Uwe Klaus Zettl

**Affiliations:** 1Department of Extracorporeal Therapy Systems, Fraunhofer Institute for Cell Therapy and Immunology IZI, 18057 Rostock, Germany; 2Division of Nephrology, Center for Internal Medicine, Rostock University Medical Center, 18057 Rostock, Germany; 3Division of Neuroimmunology, Department of Neurology, Rostock University Medical Center, 18147 Rostock, Germany

**Keywords:** immunoadsorption, multiple sclerosis, miniaturized adsorbers, neutrophil migration, blood–brain barrier

## Abstract

Immunoadsorption (IA) has proven to be clinically effective in the treatment of steroid-refractory multiple sclerosis (MS) relapses, but its mechanism of action remains unclear. We used miniaturized adsorber devices with a tryptophan-immobilized polyvinyl alcohol (PVA) gel sorbent to mimic the IA treatment of patients with MS in vitro. The plasma was screened before and after adsorption with regard to disease-specific mediators, and the effect of the IA treatment on the migration of neutrophils and the integrity of the endothelial cell barrier was tested in cell-based models. The in vitro IA treatment with miniaturized adsorbers resulted in reduced plasma levels of cytokines and chemokines. We also found a reduced migration of neutrophils towards patient plasma treated with the adsorbers. Furthermore, the IA-treated plasma had a positive effect on the endothelial cell barrier’s integrity in the cell culture model. Our findings suggest that IA results in a reduced infiltration of cells into the central nervous system by reducing leukocyte transmigration and preventing blood–brain barrier breakdown. This novel approach of performing in vitro blood purification therapies on actual patient samples with miniaturized adsorbers and testing their effects in cell-based assays that investigate specific hypotheses of the pathophysiology provides a promising platform for elucidating the mechanisms of action of those therapies in various diseases.

## 1. Introduction

Multiple sclerosis (MS) is a chronic inflammatory disease of the central nervous system (CNS). All its clinical manifestations exhibit demyelination in the white and gray matter and axonal and neuronal degeneration [1,2]. The exact cause of the onset of MS is not known, but a variety of possible factors have been identified that contribute to susceptibility for MS development. Low exposure to ultraviolet B light, vitamin D deficiency, smoking, obesity, and infection with the Epstein–Barr virus in combination with individual genetic predispositions are important factors that contribute to the development of MS [3,4]. Two main explanatory models are discussed among experts in the pathophysiology of MS, namely, the outside-in and the inside-out paradigms. The outside-in paradigm postulates an autoimmune attack against myelin originating in the periphery as the fundamental cause for the development of MS, whereas the inside-out paradigm proposes that a primary degenerative process in the CNS triggers secondary autoimmune reactions against myelin debris [5]. In a healthy subject, the blood–brain barrier (BBB) restricts the movement of soluble mediators and immune cells from the periphery into the CNS. In MS, however, leukocyte entry into the CNS is an early event. Relapses are especially characterized by an increased infiltration of peripheral immune cells across the BBB. The immune cells of subjects with MS express inflammatory cytokines, reactive oxygen species (ROS), and enzymes that can facilitate leukocyte migration to the CNS by influencing BBB function, either directly or indirectly [6]. Whether BBB dysfunction precedes immune cell infiltration or is the consequence of perivascular leukocyte accumulation remains enigmatic, but leukocyte migration modifies BBB permeability.

Immunoadsorption (IA) is clinically well-established in the treatment of steroid-refractory MS relapses [7,8,9]. However, the mechanism of action of IA in patients with MS is still unknown [8]. Generally, IA is also applied in various other (neuro)immunological diseases, e.g., systemic lupus erythematosus (SLE), neuromyelitis optica (NMO), and Guillain–Barré syndrome (GBS). The main rationale of this therapy is to remove certain harmful molecules from the circulatory system to regain homeostasis. However, the exact biological and physiological effects of IA on the human organism are much more complex and poorly understood. To explore the mechanism of action of IA and other blood purification therapies like hemoperfusion in different diseases, it is important to investigate them on the molecular and cellular level using controlled model environments that mimic clinical reality as closely as possible. This requires using sample material from the patients concerned. Since these sample materials are only available in small quantities, it is necessary to simulate extracorporeal therapies on a miniaturized scale.

Previous studies have used miniaturized adsorbers to investigate the effectiveness of blood purification therapies with regard to the removal of certain mediators in in vitro or animal models [10,11,12]. However, these studies either focused on the mechanistic removal of antibodies or other mediators that were assumed to be relevant for the pathophysiology of the respective disease or used the miniaturized adsorbers in animal models that had limited applicability to the human situation and were too complex to allow direct conclusions on the actual mechanism of action of the blood purification therapy in question.

Here, we present an experimental approach that uses miniaturized adsorbers to treat patient material and test disease-specific hypotheses for pathophysiology using cell-based assays with IA in MS as an example. The objectives of the present study are to investigate the effect of an in vitro IA model using a commercially available sorbent (tryptophan-immobilized polyvinyl alcohol gel) in miniaturized adsorbers on the levels of disease-relevant target molecules in the plasma of patients with MS and examine the influence of the patients’ plasma before and after IA treatment on immune cell migration and endothelial cell barrier integrity using in vitro cell assays (see study design in Figure 1).

## 2. Results

### 2.1. Removal of Selected Target Molecules by In Vitro IA

#### 2.1.1. Cytokines and Chemokines

We measured fifteen cytokines and four chemokines in the plasma samples of patients with MS and healthy controls (see Appendix A for the results of the validation experiments which compare the performance of miniaturized adsorbers with clinic-size Immusorba TR-350 devices). We report here a subset of the cytokine data for which the IA treatment showed the most pronounced effects (see Appendix B for the complete cytokine profile). All the treated plasma samples (except for one healthy control and one patient with MS) showed lower cytokine concentrations compared to the samples taken before IA treatment. In the healthy control plasma, the selected cytokines showed no significant changes when comparing the levels before and after IA. However, the removal of IL-12, IL-15, IL-21, IL-23, and IL-33 (see Appendix B Figure A4) from the plasma of patients with MS was more pronounced and, for IL-12 and IL-15, statistically significant (Figure 2).

We examined the removal of the chemokines CCL5, CCL20, CXCL8, and CXCL12 (Figure 3) using the miniaturized adsorbers. The IA treatment reduced the concentration significantly for CCL5 in the healthy control samples and CXCL12 in the samples of patients with MS. The remaining groups did not show statistical significance in the IA-treated samples compared to the untreated samples.

#### 2.1.2. Complement Factors

The in vitro IA treatment increased the concentration of complement factor C3a significantly in the plasma samples of patients with MS but not in those of the healthy controls (Figure 4a). Moreover, the results revealed a statistically significant difference in the C3a concentration between the IA-treated MS plasma and the IA-treated healthy control plasma samples. The concentration of complement factor C5a increased significantly in the plasma samples of both the healthy controls and the patients with MS after the in vitro IA treatment (Figure 4b).

#### 2.1.3. IgG and IgG Subclasses 1–4

The in vitro IA treatment significantly removed IgG3 in both the control and the patients with MS groups but did not affect the concentration values of the whole IgG, IgG1, 2, and 4 (Figure 5).

### 2.2. Effect of In Vitro IA Treatment on Neutrophil Migration

The percentage of phagocytosis-positive neutrophils (Figure 6a) induced by incubation with the plasma of healthy controls and patients with MS was unaffected by the in vitro IA treatment, but neutrophil migration towards the IA-treated healthy control plasma was significantly reduced (Figure 6b). Chemotaxis towards the IA-treated plasma of patients with MS was only slightly reduced compared to that towards the untreated plasma of patients with MS. The untreated plasma of patients with MS induced significantly less neutrophil migration than the untreated plasma of healthy controls.

### 2.3. Effect of In Vitro IA Treatment on Endothelial Cell Barrier Integrity

We examined the effect of IA-treated plasma on the integrity of the BBB using human induced pluripotent stem cell-derived endothelial cells (hiPSC-ECs). The incubation of the cells with the untreated plasma of patients with MS significantly compromised the BBB’s integrity, as shown by a markedly lower trans-endothelial electrical resistance (TEER) and a higher paracellular permeability of fluorescein isothiocyanate (FITC)-dextran (4 kDa) in the hiPSC-ECs compared to incubation with untreated healthy control plasma (Figure 7a,b). The in vitro IA treatment of the plasma of patients with MS significantly improved BBB integrity, as indicated by restored TEER and reduced leakage of FITC-dextran into the basolateral compartment.

Live–dead staining showed no differences between the number of dead hiPSC-ECs after incubation with the plasma of untreated and IA-treated healthy controls and patients with MS (Figure 8a,b). The quantification of DAPI-negative cells by FACS analysis also indicated no differences between all the groups (Figure 8c). The release of lactate dehydrogenase (LDH) as a parameter for cell stress tended to be higher after incubation with MS plasma compared to healthy plasma, but this difference did not reach statistical significance (Figure 8d; pre: *p* = 0.1976; post: *p* = 0.2034).

## 3. Discussion

The aim of this study was to explore the molecular and cellular mechanisms of action of IA in patients with MS. For this purpose, we used miniaturized adsorbers to treat plasma samples collected from patients with MS in an in vitro IA model and determined the levels of selected cytokines and chemokines, complement factors, and IgG in blood plasma before and after the IA treatment. Furthermore, we incubated isolated neutrophils from healthy donors and endothelial cells derived from hiPSCs with the different types of plasma to examine the effects of the in vitro IA treatment on neutrophil chemotaxis and epithelial barrier integrity, respectively.

From the investigated cytokine panel, we found that the IL-12, IL-15, IL-21, IL-23, and IL-33 levels were much lower in the IA-treated plasma of patients with MS, but only the decreases in the IL-12 and IL-15 levels were statistically significant. The removal of CXCL12 from the plasma of patients with MS was also significant. The lack of statistical significance for the other cytokine and chemokine levels is probably due to the high inter-individual differences in the baseline levels of the mediators that we observed in these patients. IL-12, IL-15, and CXCL12 are among the 21 biomarkers in blood that were identified in a meta-analysis by Bai et al. [13] to be consistently elevated in patients with MS. The authors suggested that it is possible that their removal may alter cellular immunity [13]. Generally, cytokines and chemokines play a key role in the recruitment of leukocytes from the blood to sites of inflammation in the CNS [14,15]. IL-12 induces the generation of cytotoxic T cells [16], whereas IL-15 is pivotal to B and T cell maturation [17], promotes the recruitment of T-cells, and has cytotoxicity-enhancing effects for NK and T cells [18,19,20,21,22]. CXCL12 is a homeostatic chemokine that induces the migration and activation of leukocytes [23,24] and is involved in inflammatory conditions by its synergistical interaction with the chemokine CXCL8 [25,26]. Furthermore, it has been shown for MS that CXCL12 promotes monocyte recruitment into the brain perivascular space, resulting in neuroinflammation [27]. Data on the removal of cytokines and chemokines during IA treatment are generally limited. Boedecker et al. [28] investigated the efficacy of IA with a tryptophan sorbent in patients with MS in a clinical study. The authors also found a significant removal of IL-12. However, in contrast to our study, they observed significant decreases in the IL-17, IL-6, TNF-α, and IFN-γ levels after IA treatment as well. Pfeuffer et al. [29] performed IA with Immusorba TR-350 in patients with MS with refractory relapses. They observed statistically significant removal of the following cytokines and chemokines: IL-6, IL-13, and IL-15 (IL-12 was not included in the panel). Like in our study, they did not detect significant changes in the levels of IL-17A, IL-17F, IL-33, TNF-α, and IFN-γ.

The levels of total IgG and IgG1 were only slightly decreased in our study after IA treatment. Earlier studies by Ohkubo et al. reported IgG adsorption rates between 18.2% [30] and 26.5% [31] for large-scale IA, when 3 L or 2 L of plasma was processed, respectively. The tryptophan sorbent that was used in our model has a high affinity for IgG3, a moderate affinity for IgG1, and a weak affinity for IgG2 and IgG4 [32]. Furthermore, the sorbent and patient plasma volume in our miniaturized IA model were scaled according to the ratio of a typical adsorber (350 mL) and a treated plasma volume of 3 L. Therefore, comparatively lower removal rates were to be expected. These factors may explain why only the levels of IgG3 were significantly reduced in both the plasma of healthy controls and that of patients with MS in our analysis.

We observed significant increases in the concentration of C3a in the plasma of patients with MS and C5a in the plasma of both healthy controls and patients with MS in response to IA treatment with the miniaturized adsorbers, suggesting complement activation. Complement activation involves the cleavage of complement factors such as C3, C4, and C5 into different fragments, including anaphylatoxins C3a, C4a, and C5a. Complement activation by IA has been reported in the literature for different ligands and sorbent materials [33,34,35,36,37,38,39,40]. In several of these studies, decreases in C3 and/or C4 levels were described [34,35,37,40], which were attributed to the cleavage of these factors during complement activation and/or removal by adsorption. Increases in C3a and/or C5a were also described in several studies [35,36,37,38,39]. With regard to tryptophan- and phenylalanine-immobilized PVA gel, the literature on anaphylatoxin generation is comparatively sparse and the statements of the authors are partly ambiguous, particularly for C3a. While Ota et al. [36] and Shiga et al. [39] observed marked increases in C3a and C5a and Fadul et al. [37] reported slight increases in C3a in the treatment of patients with different (neuroimmunological) diseases, which confirms the observations of the present study, Grob et al. [35] observed an increase in C5a but a decrease in C3a in patients with myasthenia gravis, and Palm et al. [34] did not observe an influence on the C3a levels in an in vitro setup. The reasons for this slightly unclear study situation could lie in the heterogeneity of the patients examined, which was also observed in the present study. For instance, the slight and non-significant increase in C3a in the plasma of healthy controls in comparison to the plasma of patients with MS may be due to lower levels of the precursor protein in the healthy controls. In agreement with this, significantly higher plasma levels of C3 were observed in patients with MS compared with controls (1454 mg/L vs. 1263 mg/L) by Ingram et al. [41]. However, the clinical relevance of the complement activation capacity of IA still remains unclear [42].

MS affects the BBB and plays a significant role in disease progression. Immune reactions in MS cause inflammation, which increases the BBB’s permeability and fuels cerebral inflammatory processes [43]. For this reason, we examined the cell barrier’s integrity in a model using brain endothelial cells derived from human induced pluripotent stem cells that closely mimic the BBB [44]. We found that IA-treated MS plasma had a significantly less detrimental effect on barrier integrity than untreated plasma. Moreover, the chemotaxis assay showed a slight trend of reduced neutrophil migration towards the IA-treated plasma of patients with MS. This reduced neutrophil migration suggests that these cells are less likely to infiltrate the brains of patients with MS when cytokines and chemokines are elevated in cerebrospinal fluid (CSF) [13]. Seven patients with MS whose plasma was treated in vitro with the miniaturized IA model had received seven different disease-modifying drugs (DMDs)—alemtuzumab, natalizumab, teriflunomide, interferon beta-1a, interferon beta-1b, glatiramer acetate, and fingolimod—for varying durations prior to blood collection. DMDs primarily target neuroinflammation [45] by either depleting (e.g., cladribine, alemtuzumab, ocrelizumab) or inhibiting (e.g., teriflunomide, interferon beta) peripheral immune cells, by sequestering them in lymph nodes (e.g., fingolimod, siponimod), or by directly blocking their ability to cross the BBB (natalizumab) [45,46]. Moreover, in vitro studies have demonstrated that, in addition to their in vivo mode of action, alemtuzumab [47], fingolimod [48], and interferon beta [49,50,51,52] affect the transmigration of immune cells across the BBB, and glatiramer acetate modulates immune cell reactivity [53]. Furthermore, a recent animal study on teriflunomide revealed that it not only inhibits immune cells but also enhances BBB integrity [54]. Therefore, it is possible that residual DMDs in the patients’ plasma impaired the cell assays in the present study.

Besides the infiltration of lymphocytes and monocytes into the CNS, neutrophils are also discussed to contribute to the pathophysiology of neuroinflammatory diseases [55,56]. The prevention of BBB impairment and the reduction in neutrophil migration could result from the removal of potentially harmful mediators such as cytokines and chemokines, as previously discussed. Regarding the clinical effect, our results indicate that the BBB is less susceptible to penetration by potentially harmful cells and mediators and that IA reduces the migration of immune cells, potentially providing a protective effect for the brain of patients with MS who receive IA for relapse treatment.

The novelty of our experimental approach lies in the in vitro treatment of patient sample material with miniaturized adsorbers and subsequent analyses with molecular and cell-based assays. This allows researchers to specifically test hypotheses regarding the mechanisms of action of blood purification procedures in various diseases under controlled laboratory conditions in vitro.

When interpreting the data, some limitations of the in vitro IA model should be acknowledged. Since only plasma samples were treated in vitro, the potential effects of IA on immune cells could not be evaluated. For instance, previous clinical studies have demonstrated that IA stimulates the secretion of IL-10 by immune cells in patients with neurological autoimmune diseases, leading to a less pronounced decrease [28] compared with other cytokines or even an increase [57] in IL-10 plasma levels. It has been proposed that this shift in the ratio of pro-inflammatory and anti-inflammatory cytokines—especially IL-10—may contribute to the clinical response to IA treatment [28]. Furthermore, our model did not account for the transfer of disease-relevant molecules and mediators from tissue and other compartments into the circulatory system, which might, in vivo, also affect immune cell migration and cell barrier integrity. IA treatments are generally performed after separating the plasma of patients from the cellular blood components using a plasma separator. Fadul et al. [37] demonstrated that plasma separation itself has a considerable impact on complement and immune cell activation [37]. Because of the limited blood that was available from the patients in our study, it was not feasible to perform plasma separation in our miniaturized IA model, which typically requires additional extracorporeal volume. The interpretation of the data was further complicated by the relatively low number of subjects in this pilot study, the high inter-individual variability in the target molecules’ levels, and the different disease and medication histories. To increase the validity and reliability of the data, more in vitro IA treatments of plasma of patients with MS are needed. Analyses of a broader spectrum of disease-relevant molecules could further help researchers to obtain a more comprehensive picture of the effect of IA at the molecular level. Testing clinical samples of patients with MS before and after IA treatment on immune cell migration and endothelial cell barrier integrity could also enhance the validity of the results.

Nevertheless, our experiments indicate that IA might induce a shift in the levels of cytokines in blood plasma and that it reduces the migration of autoreactive immune cells through the BBB. The removal of immune mediators and the stabilization of endothelial cell barrier integrity may, in part, explain the therapeutic effect of IA in steroid-refractory MS relapses. This approach can be applied to screen different sorbents and their effects on blood purification treatments in MS or other diseases, such as rheumatoid arthritis, systemic lupus erythematosus, or Guillain–Barré syndrome. This may help clarify the mechanisms of action of blood purification therapies on MS and other (neuroinflammatory) diseases and improve them.

## 4. Materials and Methods

### 4.1. Collection of Plasma Samples from Patients with Multiple Sclerosis and Healthy Controls

Blood samples from patients with MS were collected in the Department of Neurology of the Rostock University Medical Center. Blood for the control group was collected from healthy volunteers. The clinicodemographic data of the study probands are presented in Table 1. From each proband, 30 mL of blood was collected in sodium heparin monovettes. The blood samples were centrifuged immediately after sampling, and the plasma was carefully collected and stored at −80 °C until treatment with the miniaturized adsorbers in the in vitro IA model.

### 4.2. Processing of Plasma in the Miniaturized Immunoadsorption Model

We used a tryptophan-immobilized polyvinyl alcohol (PVA) gel from the Immusorba TR-350 (Asahi Kasei Medical, Tokyo, Japan) as a sorbent in the miniaturized adsorbers (Figure 9a). The IA model simulates clinical treatment by considering the ratio of sorbent volume (350 mL in the clinical adsorber) and the treated patient plasma volume, which is approximately 3 L for a standardized patient with a blood volume of 5 L and a hematocrit of 40%. This ratio is approximately 8.6 for the Immusorba TR-350. Our miniaturized adsorbers contained 0.6 mL of the sorbent. Accordingly, the plasma volume to be processed by the adsorber was 0.6 mL × 8.6 ≈ 5.2 mL. The typical plasma flow rate in the application of the clinical adsorber is 20 mL/min, which results in a treatment time of 150 min for one patient plasma volume. In our IA model, this corresponds to a plasma flow rate of 5.2 mL/150 min ≈ 0.034 mL/min. The test setup was placed in a warming cabinet set to a temperature of 37 °C (Figure 9b). A total of 4 mL of untreated patient plasma (pre-adsorption plasma) was kept in separate tubes in the warming cabinet over the duration of the treatment to consider the potential influence of prolonged sample warming on the measurements. The IA-treated plasma (post-adsorption plasma) was collected in separate tubes. After completion of the IA treatment, pre- and post-adsorption plasma samples were immediately stored at −80 °C until analysis. The IA model was validated in comparative studies with a large-scale model (see Appendix A).

### 4.3. Cytokine and Chemokine Analyses

Cytokine and chemokine plasma concentrations were determined using a commercially available bead-based multiplex immunoassay (LegendPlex; BioLegend, San Diego, CA, USA), which operates on the same principle as a sandwich immunoassay. The fluorescence intensity of each bead was measured with a flow cytometer (MACSQuant Analyzer 16; Miltenyi Biotec, Bergisch Gladbach, Germany). The concentration of each analyte was determined using a standard curve generated with the same assay.

### 4.4. Complement Factor Analyses

The concentrations of complement factors C3a and C5a were determined with sandwich enzyme-linked immunosorbent assays (Complement C3a Human ELISA Kit, Complement C5a Human ELISA Kit; Thermo Fisher Scientific, Waltham, MA, USA).

### 4.5. Antibody Analyses

Immunoglobulin G (IgG) and IgG subclasses 1–4 concentrations were determined with a nephelometer (Atellica NEPH 630 System; Siemens Healthineers, Erlangen, Germany). Measurements were conducted according to the specification of the clinically validated assays.

### 4.6. Investigation of Neutrophil Migration

A transwell migration assay was performed to quantify cell movement toward a chemoattractant. For this purpose, human neutrophils were isolated from human buffy coats (Transfusion Medicine, Rostock University Medical Center, Rostock, Germany) with a whole blood neutrophil isolation kit (MACSxpress; Miltenyi Biotec). Plasma medium (with or without chemoattractant) and patient plasma samples (before and after treatment) were added in the bottom wells of a 24-well plate. Transwell inserts were placed into each well, and 3 × 10^5^ isolated neutrophils were added to the apical chamber. The plate was incubated for 30 min at 37 °C in 5% CO_2_. Non-migrated cells where removed from the apical transwell chamber. Cells that migrated towards the transwell membrane in the basal chamber were harvested and counted using a flow cytometer (MACSQuant Analyzer 16; Miltenyi Biotec).

### 4.7. Investigation of Neutrophil Phagocytosis

For the phagocytosis assay, purified human neutrophils isolated from buffy coats (see Section 4.6) were seeded into a 96-well plate (1 × 10^5^). Patient plasma samples (before and after treatment) were added to the cells, and the plate was incubated for 1 h at 37 °C with 5% CO_2_. Subsequently, the plate was centrifuged at 300× *g* for 5 min at room temperature, and the supernatant was removed. The cells were resuspended in 100 µL RPMI cell culture media, and pHrodo™ Deep Red E. coli BioParticles Conjugate for Phagocytosis (Thermo Fisher Scientific) was added to the cells (250 µg/mL). The cells were incubated for 1 h at 37 °C with 5% CO_2_, and fluorescence was detected with a flow cytometer (MACSQuant Analyzer 16; Miltenyi Biotec).

### 4.8. Endothelial Cell Barrier Integrity Analyses

#### 4.8.1. Cell Barrier Model

To investigate the influence of patient plasma before and after treatment with the IA model on the endothelial barrier’s integrity, human induced pluripotent stem cells (hiPSC; differentiated according to Lippmann et al. [44,58]) were used. The cells were grown to confluence on transwell membrane inserts that had been coated before with collagen and fibronectin. The cells were placed in the transwell system and cultured for one day to form a functional monolayer. On day two, the cells were incubated with the untreated and IA-treated plasma of patients with MS for 24 h at 37 °C in a 5% CO_2_ atmosphere. Plasma from healthy donors served as the control.

#### 4.8.2. Transepithelial Electrical Resistance

Trans-endothelial electrical resistance (TEER) is generally used to assess the integrity and barrier function of epithelial cell layers found in epithelial tissues and cell culture models. Here, it was used to assess the integrity of the endothelial hiPSC monolayer using a voltohmmeter (Millicell-ERS; Merck Millipore, Burlington, MA, USA). TEER is affected by the tight junctions between adjacent cells, which form a barrier which restricts the passage of ions and molecules between cells. A higher resistance indicates a more intact and functioning epithelial barrier. The TEER values were measured before and after stimulation of the endothelial hiPSC with the different plasma samples. A transwell insert without cells served as the blank.

#### 4.8.3. Paracellular Permeability Assay

The fluorescein isothiocyanate (FITC)-dextran assay is a commonly used method to evaluate the integrity and permeability of endothelial barriers, especially in in vitro cell culture models. FITC-dextran (4 kDa) is a fluorescence-labeled form of dextran, a large polysaccharide molecule. FITC-dextran was added to endothelial cells in the upper compartment (apical side) of the cell culture system. The integrity of the barrier was assessed by fluorescence measurement in the lower compartment (basolateral side) after 2 h of incubation using a multiplate reader (CLARIOstar Plus; BMG Labtech, Ortenberg, Germany). If the endothelial barrier was intact and had a low permeability, the FITC-dextran molecules remained predominantly in the upper compartment, indicating limited passage of the dye through the barrier. In contrast, if the barrier was damaged or exhibited increased permeability, the FITC-dextran molecules passed through the barrier and were detected in the lower compartment. We calculated the apparent permeability coefficient (Papp) for FITC-dextran as a measure for the integrity of the endothelial cell barrier.

### 4.9. Endothelial Cell Viability Assessment

#### 4.9.1. Live–Dead Staining

The LIVE/DEAD™ Viability/Cytotoxicity Assay Kit for mammalian cells (Invitrogen/Thermo Fisher Scientific, Waltham, MA, USA) is utilized for the fluorescence microscopic evaluation of cell viability, distinguishing between live (calcein-AM, 2 µM) and dead (ethidium homodimer-1, 4 µM) cells. Following a 15 min incubation in the well under standard conditions in the incubator, the fluorescence of the cells was qualitatively assessed using a fluorescence microscope (Eclipse Ti-E microscope; Nikon, Tokyo, Japan) with GFP (457–487 nm) and Texas Red (542–582 nm) filters and documented photographically.

#### 4.9.2. Lactate Dehydrogenase in Cell Culture Supernatant

After 24 h of incubation, the cell culture supernatants were analyzed by measuring lactate dehydrogenase (LDH). A volume of 120 µL of cell culture medium supernatant/plasma was assessed through photometric measurement, detecting the change in absorbance at 340 nm. This was carried out using an automated chemistry analyzer (Cobas Mira; Roche, Mannheim, Germany) following the optimized standard method established by the German Society for Clinical Chemistry (DGKC) [59].

#### 4.9.3. Cell Viability

The determination of cell viability was performed using DAPI staining. The analysis was conducted in a 96-well format on the MACSQuant Analyzer 16 (Miltenyi Biotec). The viability was assessed as a percentage ratio of dead to living cells, with the vital dye DAPI added in a 1:100 ratio just before measurement.

### 4.10. Statistics

Statistical analysis was performed with GraphPad Prism 6.0 using the *t*-test. The paired *t*-test was used for within-group comparisons (e.g., healthy pre vs. post), labeled with (*), whereas the two-sample (unpaired) *t*-test was used for between-groups comparisons (healthy vs. MS), labeled with (#). The results are expressed as the mean + standard error of the mean (SEM). The significance level was set to alpha = 0.05, without correction for multiple testing as all analyses were exploratory in nature.

## Figures and Tables

**Figure 1 ijms-25-02590-f001:**
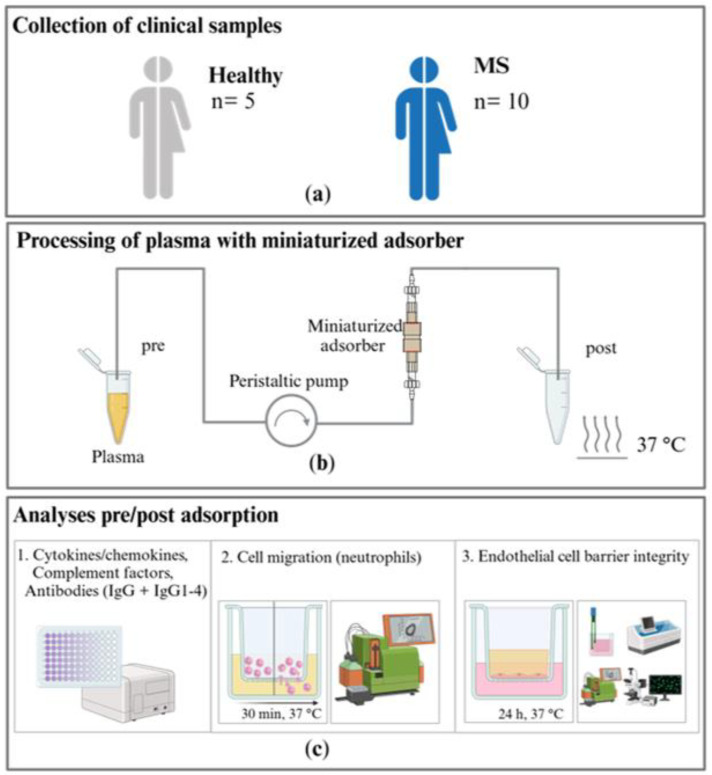
Study design: (**a**) samples from patients with multiple sclerosis (MS) and a healthy control group are collected and (**b**) treated in an in vitro immunoadsorption (IA) model with miniaturized adsorbers at a 37 °C ambient temperature in a warming cabinet. (**c**) 1. Cytokine and chemokine, complement factor, and antibody levels are analyzed in the untreated and treated plasma samples. The effects of the untreated and treated plasma on 2. the migration of neutrophils and 3. the integrity of the endothelial cell barrier are investigated.

**Figure 2 ijms-25-02590-f002:**
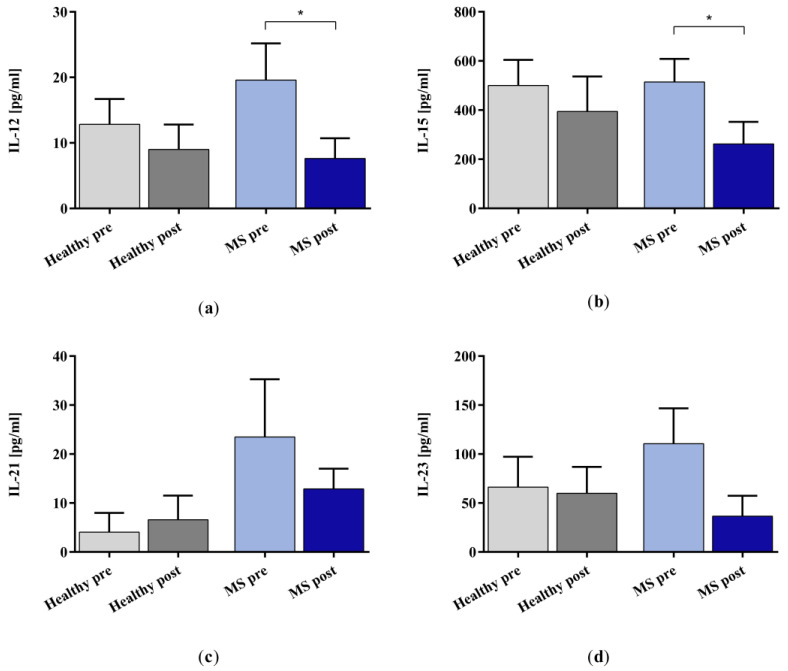
Cytokine (**a**) IL-12, (**b**) IL-15, (**c**) IL-21, and (**d**) IL-23 levels in the plasma samples of healthy controls (grey bars) and patients with MS (blue bars) before (lighter bars) and after (darker bars) in vitro IA treatment with miniaturized adsorbers. Data are shown as mean + SEM. * *p* < 0.05.

**Figure 3 ijms-25-02590-f003:**
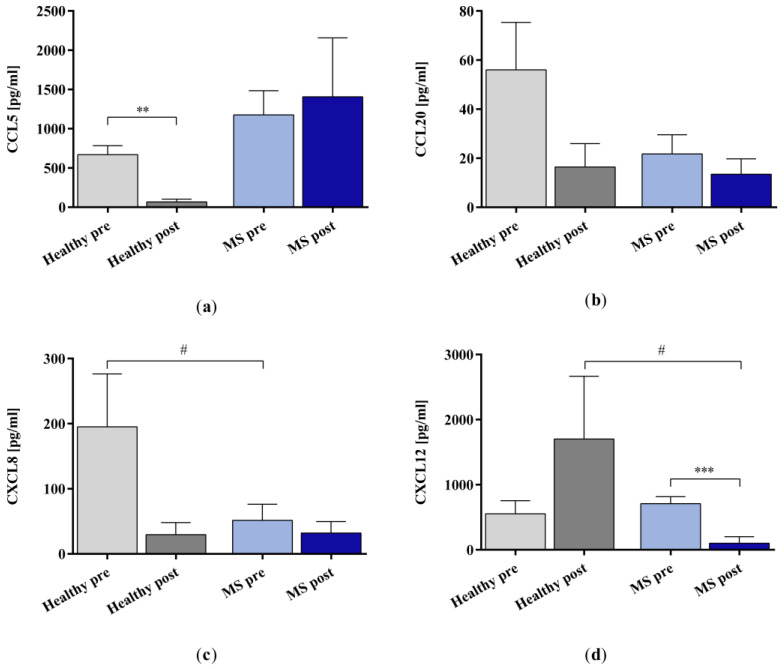
The concentrations of chemokines (**a**) CCL5, (**b**) CCL20, (**c**) CXCL8, and (**d**) CXCL12 in the plasma samples of healthy controls and patients with MS before and after in vitro IA treatment with miniaturized adsorbers. Data are shown as mean + SEM. # *p* < 0.05, ** *p* < 0.01, *** *p* < 0.001.

**Figure 4 ijms-25-02590-f004:**
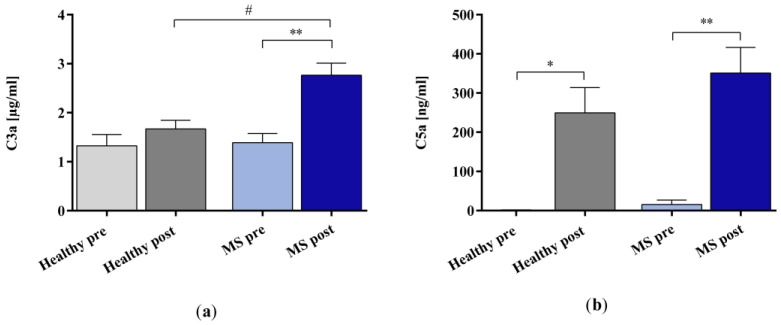
The concentrations of complement factors (**a**) C3a and (**b**) C5a in the plasma samples of healthy controls and patients with MS before and after in vitro IA treatment with miniaturized adsorbers. Data are shown as mean + SEM. # *p* < 0.05, * *p* < 0.05, ** *p* < 0.01.

**Figure 5 ijms-25-02590-f005:**
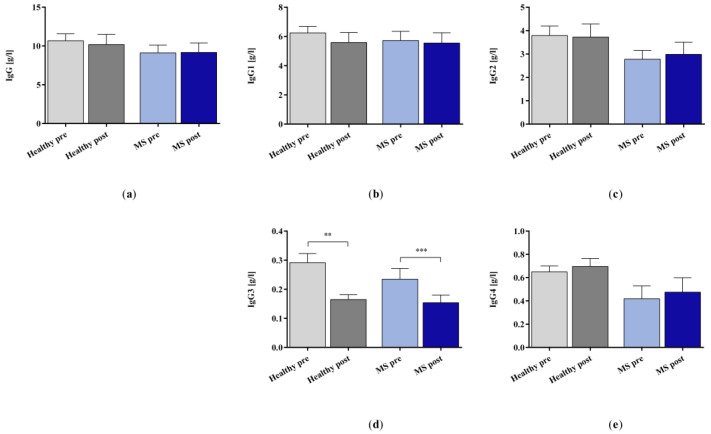
(**a**) IgG and (**b**–**e**) IgG subclasses 1–4 in the plasma samples of healthy controls and patients with MS before and after in vitro IA treatment with miniaturized adsorbers. Data are shown as mean + SEM. ** *p* < 0.01, *** *p* < 0.001.

**Figure 6 ijms-25-02590-f006:**
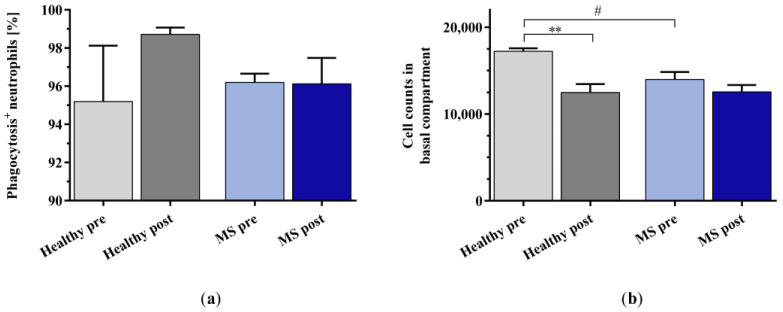
(**a**) Percentage of phagocytosis-positive neutrophils and (**b**) cell counts of buffy coat (BC)-derived neutrophils in the basal compartment of the transwell assays as a result of chemotaxis after stimulation with plasma samples of healthy controls and patients with MS before and after in vitro IA treatment. Data are shown as mean + SEM. # *p* < 0.05, ** *p* < 0.01.

**Figure 7 ijms-25-02590-f007:**
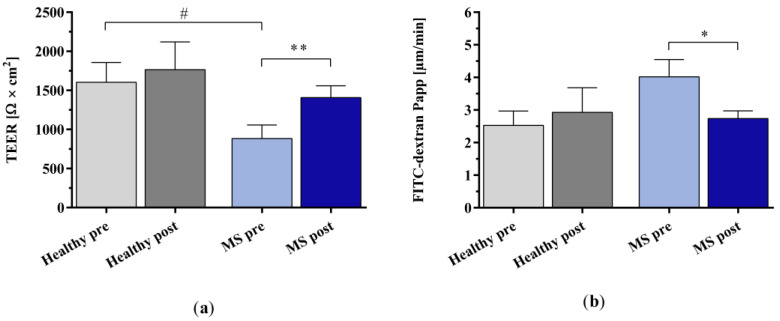
Analysis of endothelial cell barrier integrity using human induced pluripotent stem cell-derived endothelial cells (hiPSC-ECs). (**a**) Trans-endothelial electrical resistance (TEER) and (**b**) apparent permeability (Papp) of fluorescein isothiocyanate (FITC)-dextran due to paracellular passage after incubation with the plasma of healthy controls and patients with MS before and after in vitro IA treatment. Data are shown as mean + SEM. # *p* < 0.05, * *p* < 0.05, ** *p* < 0.01.

**Figure 8 ijms-25-02590-f008:**
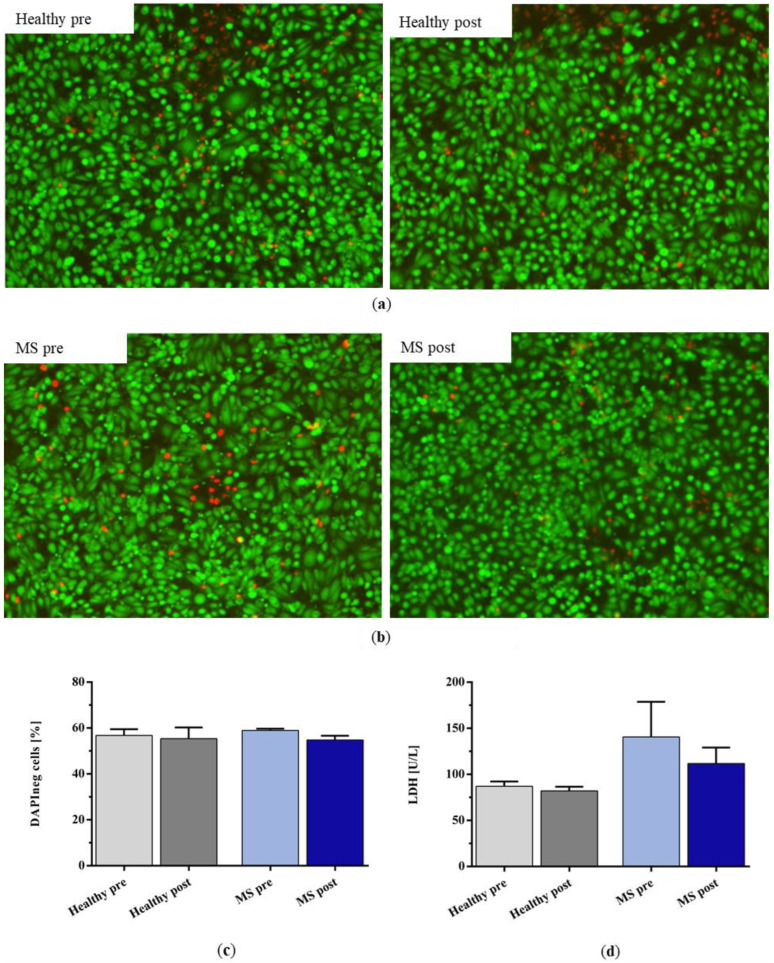
Live–dead staining (green for live cells, red for dead cells) of hiPSC-ECs after incubation with (**a**) healthy control plasma and (**b**) plasma of patients with MS before and after IA treatment with miniaturized adsorbers. (**c**) Percentage of DAPI-negative cells and (**d**) lactate dehydrogenase (LDH) concentration in the supernatant of the incubated cells after incubation with healthy plasma and the plasma of patients with MS before and after IA treatment. Data are shown as mean + SEM.

**Figure 9 ijms-25-02590-f009:**
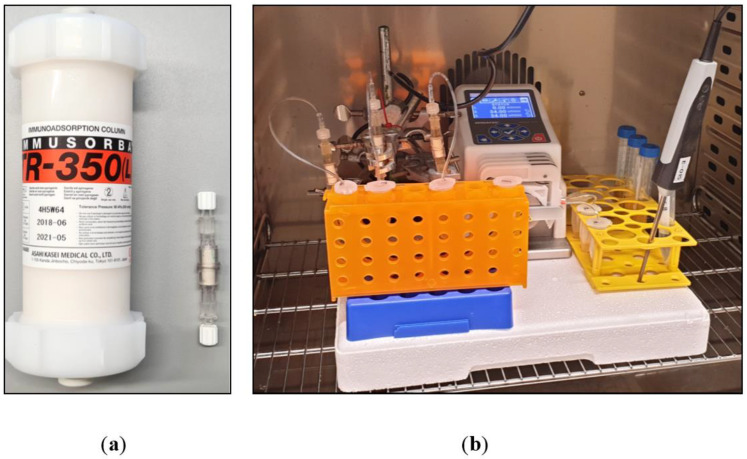
(**a**) Size comparison of Immusorba TR-350 and a miniaturized adsorber (0.6 mL sorbent volume). (**b**) Experimental setup.

**Table 1 ijms-25-02590-t001:** Clinicodemographic data.

	Healthy	MS
*n*	5	10
female	3	9
mean age in years (±SD)	32.4 (5.9)	36.5 (12.9)
mean EDSS score (±SD)	n.a.	2.3 (1.7) ^1^
mean time from MS diagnosis in years (±SD)	n.a.	8.8 (8.4)
patients on DMT	n.a.	7
patients with acute relapse	n.a.	2
median time since last relapse in months (range)	n.a.	36 (0–190)

^1^*n* = 9. DMT = disease-modifying therapy; EDSS = Expanded Disability Status Scale; MS = multiple sclerosis; *n* = number; n.a. = not applicable; and SD = standard deviation.

## Data Availability

Data are contained within the article.

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
