# Peer review of "The Utility of Miniaturized Adsorbers in Exploring the Cellular and Molecular Effects of Blood Purification: A Pilot Study with a Focus on Immunoadsorption in Multiple Sclerosis"

_ijms, 2024, doi:10.3390/ijms25052590_

Round 1

Reviewer 1 Report

Comments and Suggestions for Authors

The authors used miniaturized adsorber devices with a tryptophan-immobilized polyvinyl alcohol gel sorbent to mimic IA treatment in 10 MS patients vs. 5 healthy controls in vitro. They measured various cytokines and chemokines as well as complement factors. Moreover, they stdied phagocytosis and cell counts in the basal compartment of the transwell assays, and endothelial cell barrier integrity.  The in vitro IA treatment with miniaturized adsorbers resulted in reduced plasma levels of cytokines and chemokines. Their findings suggest that IA results in a reduced infiltration of cells into the central nervous system by reducing leukocyte transmigration and preventing blood-brain barrier breakdown. They conclude that this assay provides a promising platform for elucidating the mechanisms of action of IA in various diseases. The results are presented straight forward and are well illustrated. The discussion summarizes the main points. Just the limitations need to be expanded, and references evaluating the effect of DMTs on leukocyte migration both in-vitro and in- vivo should be added. 

Reviewer 2 Report

Comments and Suggestions for Authors

Indeed, it's an interesting article, although it should be more modest in the conclusions at the end of the discussion. They used miniaturized adsorber devices with a tryptophan-immobilized polyvinyl alcohol gel sorbent to mimic the IA treatment of MS patients in vitro. The plasma was screened before and after adsorption concerning disease-specific mediators, and the effect of the IA treatment on the migration of neutrophils and the integrity of the endothelial cell barrier was tested in cell-based models. My suggestions are few, as the paper is well-written in terms of development. I believe it would have been better to place the methods before the results; moreover, I think it's better to add 'pilot study' or something similar to the title since the sample size is very small. Finally, make the conclusions much more humble as this study needs to be replicated in a larger sample to make those affirmations

Reviewer 3 Report

Comments and Suggestions for Authors

The authors processed blood from patients and controls through a miniaturized immunoadsorption device, analyzed changes in cytokines and chemokines, and investigated the impact of post-treatment blood on the blood-brain barrier (BBB) and leukocyte migration ability. While the significance of this study is understandable, there may be fundamental issues with the experimental methodology. The following questions need to be addressed for the presentation of this study:

1. The authors need to provide a reasonable explanation (such as molecular weight or charge status) for the observed reversal in concentrations before and after treatment for IL-21, CCL20, and CXCL12 in Healthy samples compared to Patient samples. Without such clarification, it is difficult to assert the appropriateness of the experimental methods and sample sizes.

2. The authors claimed to ensure the validity of this study by comparing Miniaturized and Large scale experiments. However, for IL-12, IL-15, CCL20, and CXCL12, which showed a significant decrease in this experiment, no comparison between miniaturized and large-scale experiments was conducted. This raises concerns about the validity of using miniaturized samples and the insufficient demonstration of reflecting phenomena in vivo.

3. Regarding complement, studies by Ptak (Transfusion and Apheresis Science 32 (2005) 263–267) and Palm (doi.org/10.3109/10731199109117834) have demonstrated a decrease in vivo, and it is necessary to consider the reasons for the deviation from this study.

Round 2

Reviewer 3 Report

Comments and Suggestions for Authors

Thank you for addressing the concerns raised about your paper.